# Contacting domains segregate a lipid transporter from a solute transporter in the malarial host–parasite interface

Matthias Garten [1], Josh R. Beck [2], Robyn Roth[3], Tatyana Tenkova-Heuser[1], John Heuser[1], Eva S. Istvan [4], Christopher K. E. Bleck [5], Daniel E. Goldberg [4,6✉] & Joshua Zimmerberg [1,6✉]

The malaria parasite interfaces with its host erythrocyte (RBC) using a unique organelle, the parasitophorous vacuole (PV). The mechanism(s) are obscure by which its limiting membrane, the parasitophorous vacuolar membrane (PVM), collaborates with the parasite plasma membrane (PPM) to support the transport of proteins, lipids, nutrients, and metabolites between the cytoplasm of the parasite and the cytoplasm of the RBC. Here, we demonstrate that the PV has structure characterized by micrometer-sized regions of especially close apposition between the PVM and the PPM. To determine if these contact sites are involved in any sort of transport, we localize the PVM nutrient-permeable and protein export channel EXP2, as well as the PPM lipid transporter PfNCR1. We find that EXP2 is excluded from, but PfNCR1 is included within these regions of close apposition. We conclude that the host-parasite interface is structured to segregate those transporters of hydrophilic and hydrophobic substrates.

[1] Section on Integrative Biophysics, Eunice Kennedy Shriver National Institute of Child Health and Human Development, National Institutes of Health, Bethesda, MD 20892, USA. [2] Department of Biomedical Sciences, Iowa State University, Ames, IA 50011, USA. [3] Department of Cell Biology and Physiology, Washington University School of Medicine, St. Louis, MO 63110, USA. [4] Departments of Medicine and Molecular Microbiology, Washington University School of Medicine, St. Louis, MO 63110, USA. [5] Electron Microscopy Core Facility, National Heart, Lung and Blood Institute, National Institutes of Health, Bethesda, MD 20892, USA. [6] These authors jointly supervised this work: Daniel E. Goldberg, Joshua Zimmerberg. ✉email: dgoldberg@wustl.edu; zimmerbj@mail.nih.gov

M orbidity and mortality in malaria are due to the api-
complexan *Plasmodium spp*. replicating within the host
RBC. During its initial invasion of the erythrocyte, the
parasite invaginates the RBC plasma membrane to form the PV
as a second barrier[1,2]. The parasite must install its own, unique
transport systems in order to import and export everything it
needs for its survival and proliferation[3–5]. Understanding the
underlying transport mechanisms between the parasite and its
host is useful to identify drug targets. Yet, these crucial transport
systems remain incompletely understood for both hydrophilic
and hydrophobic substances.

A PVM channel, permeable to water-soluble nutrients like
monosaccharides and amino acids[3], is formed by the "exported
protein 2" or EXP2[6]. EXP2 also facilitates protein export, by
serving as the protein-permeant pore for the "*Plasmodium*
translocon of exported proteins" (PTEX)[7]. A number of PPM
channels, including several that are specific for particular nutri-
ents have been identified and studied[8,9]. However, it is not known
how lipidic substances are transported across the PV, since the
two limiting membranes have never been seen to connect or to
transport membrane vesicles between each other[4]. The PPM
resident protein "*Plasmodium* Niemann-Pick C1-related protein"
(PfNCR1) is essential for lipid homeostasis[10] but it is unknown
how it functions. It is not clear how the PV is organized, in order
to support the transport of such a large variety of substrates.

Here we show that the structure of the PV is built such that it
can support direct exchange of lipids across the PV space, directly
between the PPM and PVM, in regions that can be defined as
membrane contact sites (MCS)[11]. While MCS between intracel-
lular organelles are abundant[11], and cell-cell junctions are clas-
sically defined[12], very little is known about the contacts between
membranes that delimit extracellular junctions within cells, such
as those of chloroplasts and intracellular parasites. The structural
and molecular data presented here assigns a functional sig-
nificance to a macroscopic membrane domain.

## Results

**Regions of close PVM-PPM apposition exist**. Ultra-thin sections
of chemically fixed, resin-embedded parasitized red cells were
examined in the electron microscope (EM) to determine the
separation distance between PVM and PPM. Since transport
across the PV is most active in the trophozoite stage, i.e. the stage
when parasites have begun to accumulate hemozoin and grow the
fastest but have not begun to divide[13], only this stage was con-
sidered. An example image is shown in (Fig. 1a). The distribution
of separations between PVM and PPM was found to be bimodal,
indicating two distinct structural regions: regions of "close
membrane apposition", separated by ~9 nm, and regions of 'PV
lumen' with a wider mean separation of 20–40 nm. To control for
possible artifacts introduced by chemical fixation, infected red
cells were also prepared by a quick-freeze, freeze-fracture method
that preserved cells in their most lifelike state. Still, these two
distinct types of regions could be clearly discerned (Fig. 1b). In
conclusion, both regions are narrow enough that they could be
bridged by protein complexes that could interconnect the PVM
and PPM membranes, and thus are candidates for membrane
contact sites[11].

**EXP2 localizes in domains of the PVM**. A functionally char-
acterized marker of the PVM, EXP2[6], was used to investigate the
domain structure of the PVM. To test domain formation in fixed
but label-free, unmodified parasites (NF54), an immuno-
fluorescence assay using EXP2 antibodies was performed. EXP2
was found in patches, interrupted by stretches of PVM devoid of
anti-EXP2 label (Fig. 2a). To test if these regions of EXP2 were

domains in living parasites and to control for chemical fixation
artefacts, a parasite line bearing a C-terminal mNeonGreen
(mNG) fusion to the endogenous copy of EXP2 (EXP2-mNG)
was examined[14]. In these parasites, the mNG signal also was
found in continuous patches, interrupted by stretches of PVM
devoid of mNG (Fig. 2b). To quantify and visualize these
apparent domains of EXP2, we developed a simple tool: a pro-
jection of the maximum fluorescence intensity from the inside of
the parasite onto a sphere. This results in a map of the fluores-
cence signal of the periphery of the parasite (Fig. 2c–e). Both
techniques show a protein domain coverage on the order of 50%
around the parasite (Supplementary Fig. 1).

Two physically independent techniques to visualize
EXP2 spatial organization indicate its domain structure. Thus,
EXP2-mNG can serve as a robust and readily detected marker for
such PVM domains, labeling regions of protein export and
transport of small water-soluble molecules.

**EXP2 co-localizes with PV lumen**. The EM-findings of two
regions with distinct PVM-PPM separation-distances and the
light microscopy findings of domains for protein export and
nutrient import represented by EXP2 led us to hypothesize that
these features could be correlated.

To determine the distribution of the EXP2-domains relative to
PV domains with distinct lumenal space, their localization was
carefully mapped. Thus, a fluorescent label for the PV lumen was
required that could be compared with the distribution of EXP2-
mNG. To this end, mRuby3 was targeted to the PV lumen using
the signal peptide of HSP101 (PV-mRuby3)[15,16]. To verify that
PV-mRuby3 would serve as a genuine label of the PV lumen,
correlative light/electron microscopy after cryo-thin sectioning
was performed[17]. PV-mRuby3 was indeed found to localize to the
regions of wider separation between PVM and PPM, confirming
that it would serve a label for the accessible PV space (Fig. 3a,
Supplementary Fig. 2).

Co-expression and two-color imaging of EXP2-mNG and PV-
mRuby3 demonstrated a clear-cut colocalization of both labels
around the periphery of the parasite (Fig. 3b). The mean Pearson-
correlation coefficient for the analyzed sample is 0.78 [0.75, 0.80]
(mean [95% CI], $N = 38$ cells) (Fig. 3c) as a mathematical
measure for the degree of signal overlap, with numbers from −1
(perfectly anti-correlated), 0 (not correlated) to 1 (perfectly
correlated).

Thus, EXP2 distribution can be correlated with regions of the
wide, mRuby3 accessible lumen. EXP2 is bound to the PVM,
while PV-mRuby3 is detected in the PV-lumen in between the
PVM and PPM, on average 20–40 nm from the PVM. Within the
limits of optical microscopy, they colocalize. Therefore, protein
export, nutrient import, and presumably aqueous waste export
occur in the regions of the PV lumen.

**PfNCR1 anti-localizes with EXP2 and the PV lumen**. Sites of
close membrane contact are implicated in direct transfer of lipids
via intervening or included proteins that localize to those
domains[11]. In the PPM PfNCR1 has been found to be essential
for the maintenance of lipid homeostasis[10]. However, its human
homolog, Niemann-Pick C1 (hNPC1) relies on a co-factor
hNPC2, to transport lipids[18]; no such co-factor has been identi-
fied in *Plasmodium spp*., suggesting that PfNCR1 may work dif-
ferently. Thus, localizing PfNCR1 with respect to the separation-
distances of the membranes of the PV is indicative in order to
determine if PfNCR1 can be expected to transport lipids by
interacting with a soluble cofactor from the PV lumen, or whether
it interacts directly with the PVM at sites of close membrane
apposition.

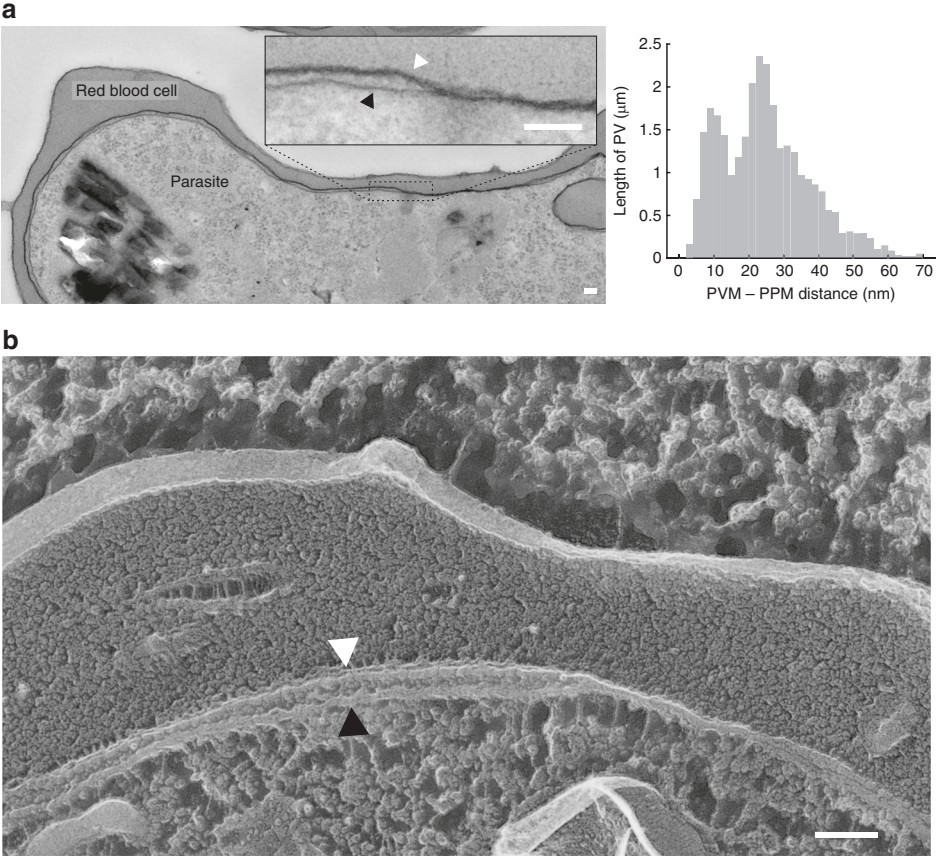

**Fig. 1 The PV exhibits two structurally distinct regions. a** (left) Thin section electron micrograph of a *Plasmodium falciparum* (NF54attb) infected red blood cell. The inset highlights the PVM and PPM. (right) Histogram of the PVM-PPM distance collected from single sections of seven parasites in six images in regions where both membranes are cut at a right angle. The histogram *y*-axis is scaled to reflect length of the sampled PV: 29.4 μm PV were sampled in total. The distribution can be fit to a two-component mixture model (see methods) with mean 1 = 8.7 nm [8.3 nm, 9.1 nm] (mean [95% CI]) and mean 2 = 26.9 nm [26.6 nm, 27.3 nm]. **b** Freeze fracture of NF54attb. A region of PV lumen is visible in the left of the image and a region of close membrane apposition is visible in the right side. **a**, **b** Scale bars: 100 nm. White arrows heads to the PVM, black arrow heads to the PPM. Source data are provided as a Source Data file.

To determine the distribution of PfNCR1 relative to the domains defined by EXP2 and the PV lumen, an endogenous EXP2-mRuby3 fusion was engineered into a parasite expressing an endogenous PfNCR1-GFP fusion protein, allowing both proteins to be monitored by live fluorescence while preserving their native timing and expression levels. Two-color imaging showed that localization of the two labels is anti-correlated around the periphery of the parasite (Fig. 4a). The mean Pearson-correlation coefficient of this sample is −0.18 [−0.08, −0.28] (mean [95% CI], N = 39 cells), giving a mathematical measure for the anti-correlation of both signals (Fig. 4b). Positive values of the coefficient are caused by small domains, approaching the resolution limit of light microscopy (~130 nm in x–y direction and ~400 nm in z) with signals co-localizing at the border of domains. In contrast, larger domains having fewer domain borders have more negative coefficients (see the sequence in Fig. 4a left to right). Anti-correlation was also observed in parasites where the PV-mRuby3 lumenal reporter was expressed in the PfNCR1-GFP background (Supplementary Fig. 3). For direct localization of PfNCR1 relative to PVM-PPM distance, the immuno-gold labeled PfNCR1-GFP dataset from Istvan et al.[10] was reanalyzed. In this dataset PfNCR1 is enriched 40.0×[14.6×, 61.4×] (mean [95% CI], N = 1118 54 nm long membrane segments) in the closely apposed regions compared to the PV regions with wider PVM-PPM distance (Supplementary Fig. 4).

One function of proteins at MCS can be maintenance of membrane distance[11]. To determine if PfNCR1 is involved in the maintenance of membrane distance, PVM-PPM distance was recorded in PfNCR1 knock-down parasites. Compared to the control, the knock-down shows large overlap of both means and confidence intervals describing membrane distance. While there was more region with larger membrane distance in the knock-down sample, this observation was found to be not significant (Supplementary Fig. 5).

From the observed anti-localization of the lumenal vs. hydrophobic transporter labels and the localization of PfNCR1 in electron micrographs, PfNCR1 is most likely localized to regions of close membrane apposition. The lack of effect of PfNCR1 knock-down on the PV structure shows that PfNCR1 seems not to be playing a major role in the regulation of membrane distance.

## Discussion

In studying how the malaria parasite modifies its protective membrane barriers to import and export materials that it needs to survive and grow, we found that the host cell–parasite interface (HPI) is a unique intercellular junction with clearly definable regions of protein composition and variable separation, consisting of a parasite vacuolar membrane, the lumen of the vacuole (that is the parasite's extracellular space), and the plasma membrane of the parasite. Different classes of molecules, hydrophobic and

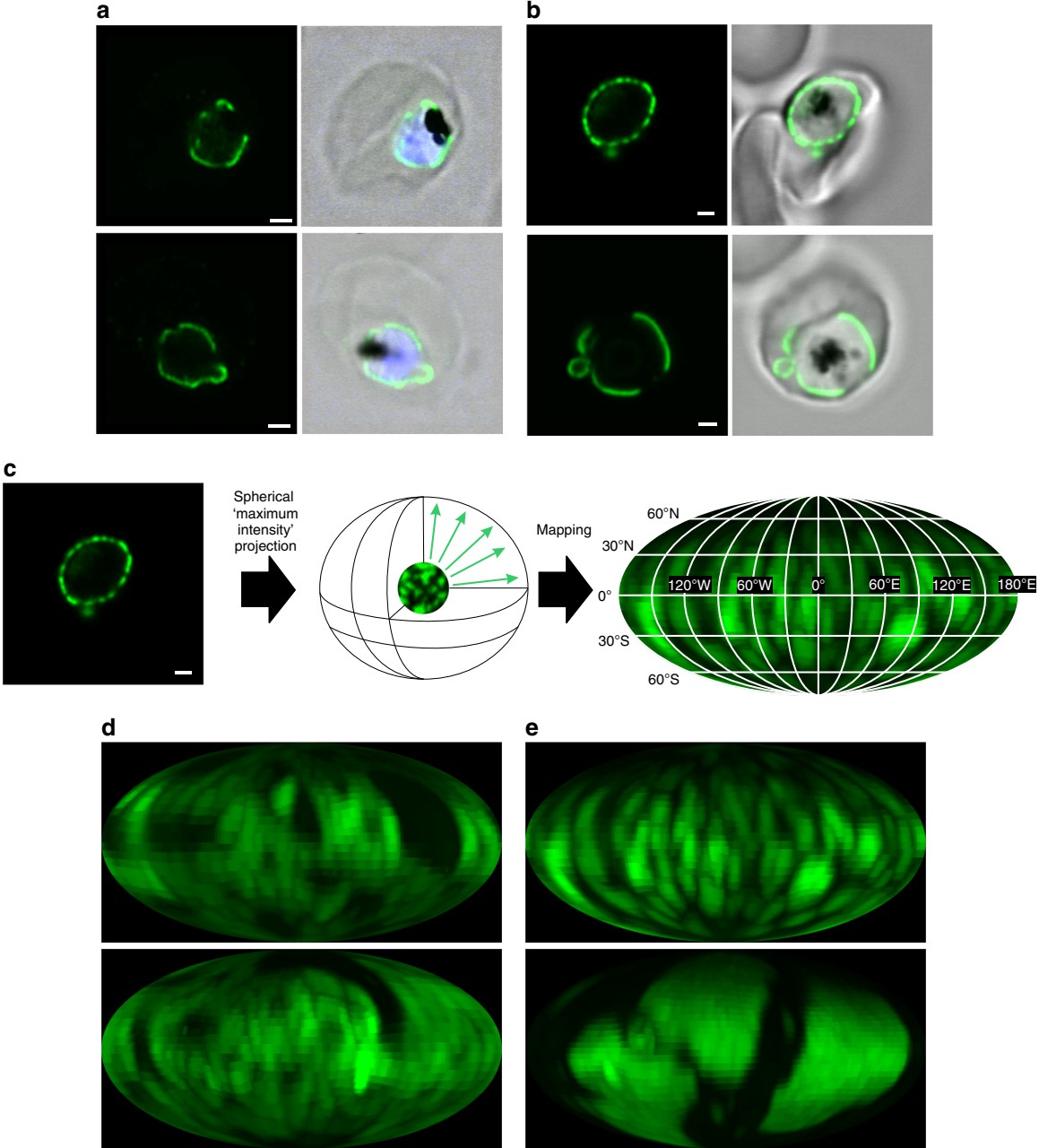

**Fig. 2 EXP2 is distributed in domains on the PVM. a** Single confocal slices of an indirect immunofluorescence assay of NF54 showing EXP2 (green), DAPI (blue) and bright field (gray). **b** Single confocal slices of a live cell image of EXP2-mNeonGreen. EXP2-mNeonGreen (green), bright field (gray). (A&B) Scale bar: 1 μm. **c** Principle of the spherical intensity projection and mapping for the 2-dimensional analysis of the 3-dimensional dataset (see methods for details). **d**, **e** Mollweide projection of the images in **a**, **b**, respectively.

hydrophilic, are transported through this one continuous, spheroidal interface with distinct regions. This segregation of function is accomplished by the creation of MCS like those between cellular organelles[19,20]. Electron microscopy resolved the PVM and the PPM, allowing quantification of their separation-distances, and confirming the visual impression that the PVM and the PPM form distinct domains, characterized by a bimodal distribution of space between the two membranes. Using two complimentary techniques, indirect immunofluorescence and live-cell microscopy of fluorescently tagged EXP2, this solute-transporter was detected in μm-sized domains that correlate spatially with domains where the PVM and the PPM are separated enough from each other for the vacuole to accumulate a

visible amount of the lumenal marker PV-mRuby3. In contrast, we found that the parasite lipid transporter PfNCR1 was specifically excluded from these regions and instead accumulated in the intervening regions of close PVM-PPM apposition. Taking all this data into account, we propose that the PV has evolved to become laterally segregated into regions for hydrophilic transport, and separate closer-contact regions for hydrophobic transport. It will be necessary to functionally characterize and localize other proteins at the HPI to further bolster this hypothesis.

The distances between the PPM and PVM for both domains could be bridged by proteins, potentially qualifying both regions as MCS[11]. However, in neither domain of the HPI were bridging proteins observed in our deep-etch EMs, nor were they observed

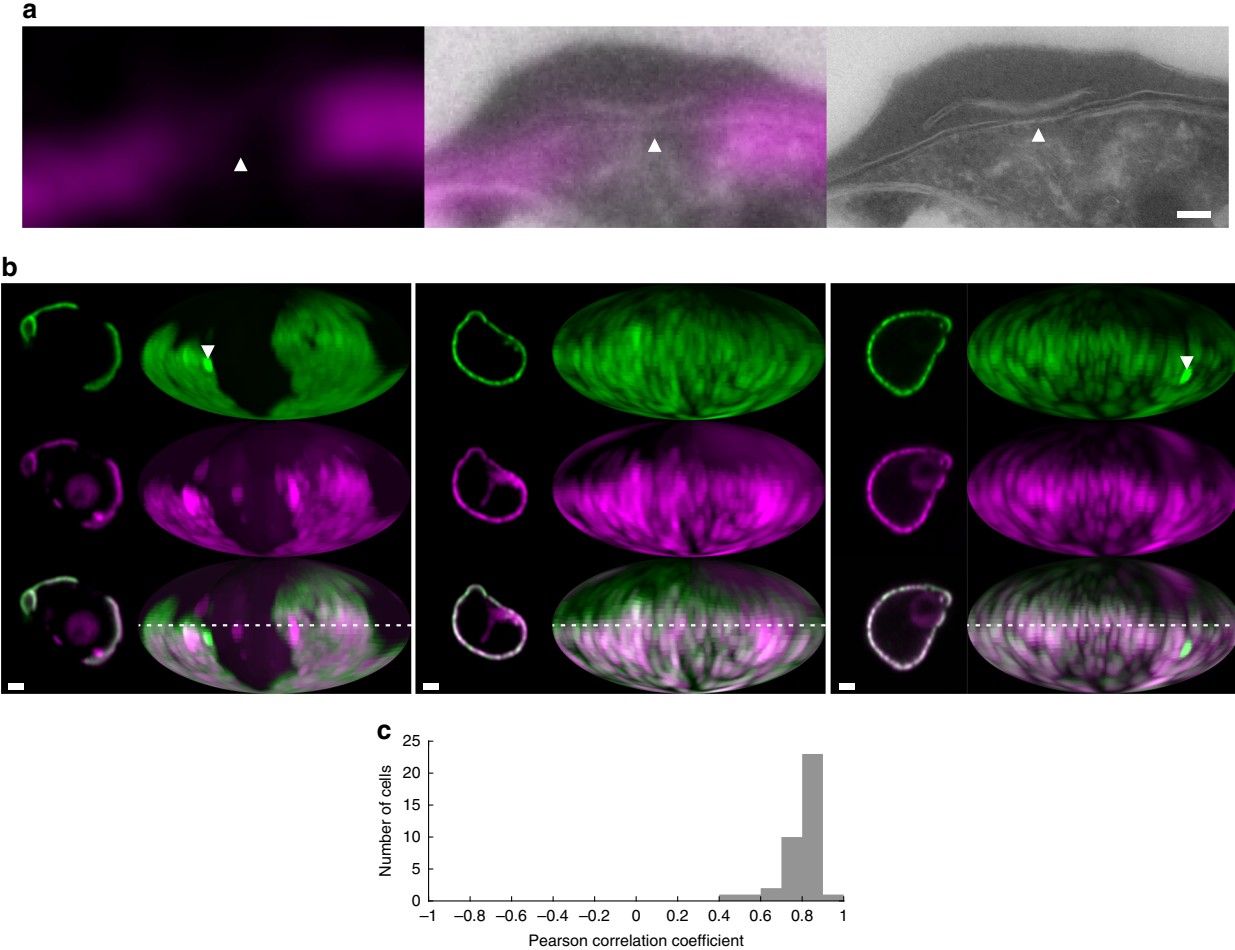

**Fig. 3 Colocalization of vacuolar space with EXP2. a** Detail of correlative light electron microscopy of EXP2-mNeonGreen—PV-mRuby3 parasites showing the presence of PV-mRuby3 in the PV lumen and absence in the apposed membrane region (arrowhead) (see Fig. SI 2 for the full image, more details and mNeonGreen channel). (magenta, left) mRuby3 signal in the Tokuyasu cryo-section. (middle) Overlay of confocal and electron microscopy image. (right) Higher resolution electron microscopy image of the region. Scale bar: 100 nm. **b** Center confocal slice and Mollweide projections of EXP2-mNeonGreen—PV-mRuby3 parasites. EXP2-mNeonGreen (green, top), PV-mRuby3 (magenta, middle), merge (bottom). Samples chosen represent examples of relatively low (0.76), average (0.80) and high (0.89) Pearson correlation coefficients of the maps. White arrowheads show tubulovesicular network[51] that has a relatively higher density of EXP2. Scale bars: 1 μm. The dotted line labels the equator, corresponds to the confocal slice. **c** Histogram of Pearson correlation coefficients of trophozoite stage EXP2-mNeonGreen—PV-mRuby3 parasites ($N = 38$ cells). Regions that extend from the parasite to form the tubulovesicular network were excluded for the correlation analysis. Source data are provided as a Source data file.

in our thin-section EMs. However, when the PV is induced to swell experimentally, e.g., when PVM protein export is conditionally impaired, leading to protein accumulation[6,21,22], it expands inhomogenously into irregular protuberances, suggesting that some sort of adhesion normally exists between the PPM and the PVM that prevents it from swelling uniformly. Still to be determined is whether these adhesions concentrate in the tightly apposed regions or the more open regions. EXP2-mNG is included in the distended regions of the PVM when protein export is impaired (Supplementary Fig. 6), indicating that the EXP2-containing domains form the less strongly connected region. Recently, EXP1, a PVM protein that colocalizes with EXP2[23,24], has been shown as important for peripheral EXP2 localization around the parasite and function of EXP2 as a nutrient-permeable channel[16,24]. It remains to be seen how EXP1 is affecting EXP2 localization and function.

In contrast, and in keeping with the detrimental effect of water on lipid transport, sites of close PVM-PPM apposition seem to be devoid of PV lumen altogether. This is an extremely close apposition, at the lower end of the membrane distances found at

organelle-organelle MSC, i.e. 10–80 nm[11]. PfNCR1, a lipid transporter, localizes to these sites of unusually close membrane apposition. While it can be inferred that lipid transport is taking place at these sites, it is unclear how PfNCR1 goes about exchanging its substrate, likely cholesterol or another lipid, with the PVM. It has been demonstrated that the large extra-membranous domain of PfNCR1 is localized in between PPM and PVM. The closely apposed membranes are within the protein's hypothesized radius[10], so it may well hand over lipids directly[25], or exchange lipids with the help of a membrane-bound cofactor. Our data suggests that PfNCR1 is not necessary for the creation of the contact sites and thus has no defining role in the PV structure. Curiously, the domain structure of the PVM, exemplified by the EXP2 distribution shown here, is quite dynamic and variable (cf., Supplementary Movie 1, demonstrating remarkable flexibility in the PV), suggesting active mechanisms of protein localization driven by active processes, e.g. cytoskeletal rearrangements coupled to the PVM by contact sites, or on-going exocytosis and solute export modifying the PV lumen. Additionally, proteins may target to their respective

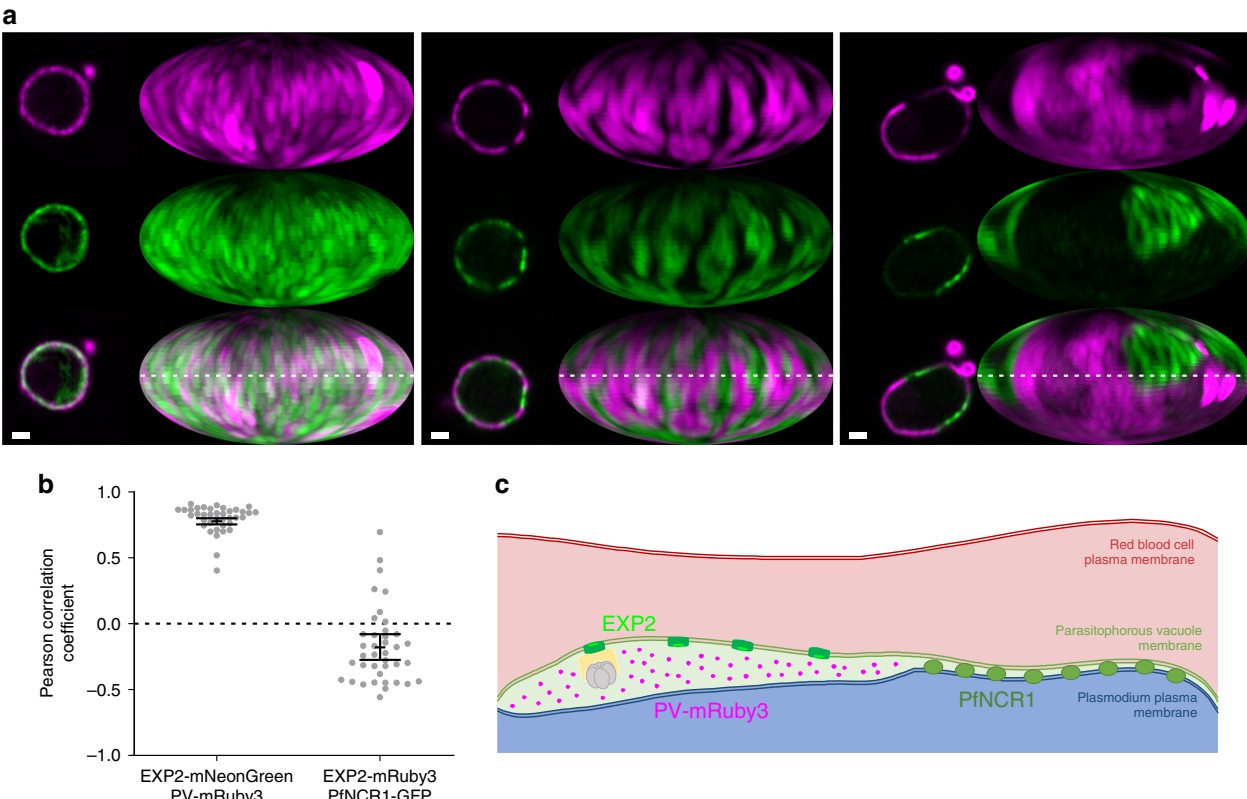

**Fig. 4 Anti-colocalization of EXP2 and PfNCR1. a** Center confocal slice and Mollweide projections of PfNCR1-GFP—EXP2-mRuby3 parasites. EXP2-mRuby3 (top, magenta), PfNCR1-GFP (middle, green), merge (bottom). Samples chosen represent examples of relatively low (0.48), average (−0.09) and high (−0.55) anti-correlation according to the Pearson correlation coefficients of the maps. The dotted line labels the equator, corresponds to the confocal slice. Scale bar: 1 μm. **b** Pearson correlation coefficients of EXP2-mNeonGreen—PV-mRuby3 parasites ($N = 38$ cells, see Fig. 3) in comparison to PfNCR1-GFP—EXP2-mRuby3 ($N = 39$ cells). Regions that extend from the parasite to form the tubulovesicular network were excluded for the correlation analysis. Bars: mean ± 95% CI. Both distributions are significantly different: $P = 1.2 \times 10^{-29}$ (t-test, two tailed). **c** Localization of EXP2 correlates with the existence of a relatively large PV lumen. The PV lumen can store proteins for export and accessory proteins to facilitate protein export. The lipid transporter PfNCR1 localizes to regions of close PPM-PVM apposition. Functionally, PfNCR1 may be able to directly contact the PVM to exchange lipids and sites of membrane apposition may be sites for the general exchange of lipophilic material, a functional hallmark of membrane contact sites. Source data are provided as a Source data file.

regions by various other mechanisms, such as interacting with structural proteins, sensing of membrane distance, as found with other contact sites[11], or protein exclusion, as found at gap junctions[12]. On longer time scales, over the parasite life cycle inside an RBC, domain arrangement and size changes (Supplementary Movie 2). While we focused on the trophozoite stage parasite, it will be insightful to learn more about the early and late PVM-PPM coordination.

The structure-function relationship described here (Fig. 4c) can potentially guide the study of other functions at the HPI, such as the inhomogeneously distributed PVM proteins noted previously[26,27]. A mechanism for the transport of PVM resident proteins from the PPM to the PVM is still lacking as PTEX is not involved in this process[28]. The regions of close apposition are a promising place to look for machinery that would allow transfer of such proteins.

The size of these domains are similar to those created in lipid phase demixing seen in model membranes[29]. However, cellular membranes are complex mixtures that for the most part fail to demix at physiological temperatures and remain almost entirely liquid-disordered[30]. Moreover, cholesterol-rich domains are not detected by surface imaging using mass spectroscopy[31]. However, molecular dynamic simulations have revealed nanoscopic microdomains of hexagonally packed saturated lipids[32] that may

correspond to the nonideal demixing revealed by FRET[33]. Recently a fungal vacuole was reported to exhibit temperature-dependent lipid phase demixing[34] but no function has yet been ascribed to these domains. The specific role of lipid asymmetry and composition, and its role in domain formation and maintenance, in the face of lipid transport, remain to be investigated for any MCS or membrane domain. The HPI offers a larger platform for these studies that may benefit both cell biological and medical investigations.

## Methods

**Cell culture**. Plasmodium falciparum was cultured in T25 Nunclon Delta closed cap culture flasks (Thermo Fisher, Waltham, MA) in RPMI 1640 supplemented with 25 mM HEPES, 0.1 mM hypoxanthine, 25 μg/ml gentamicin (all Thermo Fisher), 0.5% Albumax II (Gibco, Gaithersberg, MD), and 4.5 mg/ml glucose (MilliporeSigma, St. Louis, MO) at 37 °C in a 5% CO2, 5% O2 atmosphere at 5% hematocrit. Red blood cells were obtained from anonymized healthy donors in a protocol approved by the National Institutes of Health Institutional Review Board.

NF54 obtained through BEI Resources (MRA-1000), NIAID, NIH as part of the Human Microbiome Project. The parasite lines EXP2-mNeonGreen[14], EXP2-mNeonGreen—PV-mRuby3[15], NF54attb[35] (originally obtained from the authors of ref. [35]) were described previously.

**Molecular biology**. For generation of an endogenous EXP2-mRuby3 fusion, the blasticidin-S deaminase (BSD) cassette between SalI and BglII in the plasmid

pGDB[36] was isolated by digestion and gel purification and inserted between the same restriction sites in the plasmid pyPM2GT-EXP2-mNeonGreen[14] with a T4 Quick Ligation kit (NEB), replacing the yDHODH cassette. A synonymous mutation in the EagI site within the BSD coding sequence was then introduced to inactivate this site using a QuikChange Lightning Multi Site Directed Mutagenesis kit (Agilent, Santa Clara, CA) and the primer BSDinact (Supplementary Table 1). The mRuby3 coding sequence was then PCR amplified from plasmid pLN-HSP101-SP-mRuby3[15] with primers E2mRubyF and E2mRubyR (Supplementary Table 1) and inserted between AvrII and EagI using an In-Fusion cloning kit (clontech), replacing the mNeonGreen coding sequence and resulting in the plasmid pbBPM2GT-EXP2-mRuby3. This plasmid was linearized at AflII and co-transfected with the pAIO-EXP2-CT-gRNA[14] into a parasite line bearing a C-terminal GFP fusion to the endogenous *pfncr1* gene[10] and selection with 2.5 µg/ml blasticidin-S was applied 24 h after transfection. For expression of PV-targeted mRuby3, the plasmid pLN-HSP101-SP-mRuby3 was co-transfected with pINT[37] into the PfNCR1-GFP background and selection with 2.5 µg/ml blasticidin-S was applied 24 h after transfection.

To generate an endogenous EXP2-mNeonGreen fusion in the HSP101[DDD] background, the plasmid pyPM2GT-EXP2-mNeonGreen was co-transfected with pAIO-EXP2-CT-gRNA into NF54attB: HSP101[DDD6]. Selection was applied with 2 µM DSM1 24 h post transfection and parasites were cloned by limiting dilution when they returned from selection.

To generate a parasite line with endogenous EXP2-mNeonGreen and PTEX150-mRuby3 fusions, the mRuby3 coding sequence was PCR amplified from plasmid pLN-HSP101-SP-mRuby3[15] with primers P150mRubyF and P150mRubyR (Supplementary Table 1) and inserted between AvrII and EagI in plasmid pPM2GT-HSP101-3xFlag[6]. Homology flanks targeting the 3′ end of PTEX150 were then PCR amplified from plasmid pyPM2GT-PTEX150-3xHA-GFP11[6] with primers P150FLF and P150FLR (Supplementary Table 1) and inserted between XhoI and AvrII, resulting in the plasmid pPM2GT-PTEX150-mRuby3. This plasmid was linearized at AflII and co-transfected with pAIO-PTEX150-CT-gRNA[6] into the parasite line EXP2-mNeonGreen[14]. Selection was applied with 10 nM WR99210 24 h post transfection and clonal lines were isolated by limiting dilution, resulting in the line NF54attB::EXP2-mNeonGreen+PTEX150-mRuby3.

All primers are listed in Supplementary Table 1.

**Immunofluorescence assay.** Immunofluorescence assays (IFAs) were performed as described previously[6]. Briefly, cells taken directly from the culture and left to settle for 10 min on Concanavalin A (Vector Laboratories, Burlingame, CA) coated cover slides in culture medium at 37 °C. Excess cells were taken off with three gentle washes using 37 °C cell culture medium. Cells were fixed for 15 min at 37 °C in freshly prepared 4% paraformaldehyde, 0.0075% glutaraldehyde (both from electron microscopy sciences), in phosphate buffered saline (PBS) (Gibco). After three washes in PBS cells were permeabilized in freshly prepared PBS containing 0.2% Triton X-100 (MilliporeSigma). The sample was blocked in PBS + 3% Bovine Serum Albumin (BSA) (MilliporeSigma). Cells were incubated in primary Monoclonal antibody 7.7 (anti-EXP-2), obtained from The European Malaria Reagent Repository (http://www.malariaresearch.eu, 1:500) for 1 hr at room temperature. The sample was washed five times in PBS. The secondary antibody (Donkey-anti-mouse conjugated with Alexa Fluor 488, A21202 from Invitrogen, lot 1113537, 1:150) was applied for 20 min, then washed off 5 times with PBS. The sample was mounted in ProLong AntiFade Gold with DAPI (Invitrogen).

**Light microscopy.** Images were obtained on a Zeiss 880 with Airyscan module using a 63×1.4NA Zeiss Plan-Apochromat, 37 °C immersion oil (Zeiss, Oberkochen, DE). Images were collected using Zen black (Zeiss) in the Airyscan mode, following the programs recommendation for the optimal pixel size and slice thickness, pixel dwell times were kept at 1–2 µs. Live parasites were transferred into a hybridization chamber (HybriWell HBW20, Grace Bio-Labs, Bend, OR) for observation on the microscope[15]. The stage was heated to physiological temperature using a stage incubator (Tokai Hit INU, Fujinomiya-shi, Japan) (set temperatures: top 37 °C, stage 36 °C, objective 39 °C) to reach 35–36 °C at the coverslip. Trophozoites, i.e. non-segmented cells with hemozoin, were chosen from a bright field image. For colocalization imaging following parameters were chosen: Imaging on EXP2-mNeonGreen—PV-mRuby lines was done using a "band pass 495–550 nm + long pass 570 nm" emission filter, switching excitation at each line between 488 nm at 0.1% and 561 nm at 1% power. For experiments with GFP as fluorescent tag the line switch strategy, while minimizing movement artefacts, lead to bleed through from the mRuby3 to the GFP channel as the mRuby3 signal was very abundant and mRuby3 can be minimally excited with 488 nm making it visible alongside the relatively weak GFP signal in the GFP channel. This required switching filters after completing z-stacks in each individual channel. Parameters chosen to image PfNCR1-GFP—EXP2-mRuby3 are for the GFP channel excitation: 488 nm at 2% power, emission: "band pass 420–488 nm + band pass 495–550 nm", and the mRuby3 channel excitation: 561 nm 0.2%, emission: bandpass "495–550 nm+ long pass 570 nm". The images were processed in Zen black using Airyscan processing in automatic settings.

The whole cycle time lapse movie (Supplementary Movie 2) was made on a Zeiss LSM 800 using a 63 × 1.4NA Zeiss Plan-Apochromat equipped with a heating box to maintain 37 °C. Parasites were cultured in a gased (5% CO$_2$, 5% O$_2$) T25

tissue culture flask in which a window was cut and a cover slide was glued on using silicone, similar to the procedure outlined in ref. [38]. Light exposure was kept minimal to avoid accumulation of phototoxicity.

**Map projection and correlation analysis.** PVM, PV lumen and PPM are not resolvable from each other with a light microscope but appear as a single surface. To avoid thresholding for the correlation analysis of the light microscopy data, the 3-dimensional dataset was reduced to two dimensions projecting the maximum fluorescence intensity from the calculated center of the parasite onto a sphere in 1° intervals in all directions. This information can then be used to draw an angular map of the signal. A Mollweide projection was chosen as it represents an equal-area projection when assuming a spherical parasite. Maps obtained this way are similar to how the night sky can be represented in a map[39]. The Pearson correlation coefficient was then calculated on the maps. The maps also provide an informative visual impression of the 3D dataset in print.

Analysis was done using custom scripts in MATLAB 2018b (MathWorks, Natick MA). Briefly, the center of the cell was determined from the center of mass of a mask created from pixels in between the 1st and the 2nd level of a 2-level threshold using the "multithresh" function. The voxel was scaled according to the voxel sizes given by the microscope and the z-dimension was corrected for the refractive index mismatch between the immersion (Zeiss immersol 518 F, $n =$ 1.518) and the sample[40]. The sample index was entered as $n = 1.383$ as referenced in ref. [41]. Each voxel was then assigned an altitude and azimuth with respect to this center using the "acos" and "atan2" functions respectively. The altitude and azimuth were then used to create a mask with a 1° resolution (equivalent to 17.5 nm at a distance of 1 µm from the center). The maximum of masked intensity was then recorded as angular intensity value for the altitude and azimuth. Finally, the angular intensity values were mapped using a Mollweide projection onto a 1024 × 2048 pixel sized image. For each pixel with coordinates xi and yi corresponding angles were calculated:

$$ps = 1024;$$
$$R = ps/(2*sqrt(2));$$

$$y = ((yi - (ps/2))/(ps/2))*(R*sqrt(2));$$
$$th = asin(y/(R*sqrt(2)));$$
$$Latitude(yi,:) = real(asin((2*th + sin(2*th))/pi)*180/pi);$$

and

$$x = ((xi - ps)/ps)*(2*R*sqrt(2));$$
$$Longitude(yi,xi) = real((pi*x)/(2*R*sqrt(2)*cos(th))*180/pi);$$

Area coverage of the signal was determined by segmenting the individual maps with the 'multithresh' function for 2 levels, counting everything above the first threshold as signal.

Correlation coefficients (r) were correlated using the Pearson correlation coefficient from the intensities of channel 1 (XL) and channel 2 (YL):

$$N = length(XL);$$
$$mX = mean(XL);$$
$$mY = mean(YL);$$
$$r = (sum(XL.*YL) - N*mX*mY)/(sqrt(sum(XL.*XL) - N*mX*mX)$$
$$*sqrt(sum(YL.*YL) - N*mY*mY));$$

Regions showing the tubulovesicular network, identified as PVM extending out from the parasite into the RBC cytosol, have enriched EXP2 signal compared to the peripheral PV and were manually masked out when calculating the correlation coefficient.

The 95% confidence interval of the correlation coefficient and statistical significance was calculated in Prism (Graphpad, San Diego, CA) after a Fisher transform of the data, the resulting confidence interval was then back transformed.

**Electron microscopy on thin sections.** For the determination of the PVM-PPM distance in Fig. 1, erythrocyte cultures infected with NF54attb parasites were enriched for late stages using a LD column in a QuadroMACs magnetic separator (Miltenyi Biotech, Cologne, DE), then fixed with 2% glutaraldehyde in 100 mM NaCl, 30 mM Hepes buffer pH 7.2, and 2 mM CaCl$_2$. During a 1–2 day long aldehyde fixation, the cultures settled into soft pellets, after which they were postfixed for 30 min as pellets in 0.25% OsO$_4$ plus 0.25% potassium ferrocyanide dissolved in 0.1 M cacodylate buffer containing 2 mM CaCl$_2$. Thereafter, they were washed in 0.1 M cacodylate buffer+2 mM CaCl$_2$, and poststained by sequential 30 min treatments with 0.5% tannic acid in 0.1 M cacodylate buffer + 2 mM CaCl$_2$ followed by 0.5% uranyl acetate in 50 mM acetate buffer pH 5.2. Immediately thereafter, they were dehydrated with increasing concentrations of ethanol, and embedded in epoxy resin by standard techniques: exchange through propylene oxide, then increasing concentrations of the epoxy, and final vacuum-embedding and polymerization at 70 °C. Thereafter, semi-thin sections were made vertically

through the pellets at 0.5 μm and stained with toluidine blue, to determine optimal areas for further examination. These were then sectioned at 80 nm, stained for 5 min with 50 mM lead citrate dissolved in 0.1 M citrate buffer pH 5.2, and examined at 80 KV in a JEOL 1400 electron microscope equipped with a 4Kx6K digital camera (Advanced Microscopy Techniques Corp, Woburn). In digital versions of the final electron micrographs, PPM and PVM were segmented by hand, using ImageJ[42], drawing a line where each membrane was clearly visible. Sections where the membranes were cut obliquely were not considered. The minimal distance of the two membrane traces was determined using a MATLAB script.

From the resulting bimodal histogram, the means and relative contribution of the two PV regions were determined by a fit to a two-component mixture model. The frequency data was plotted as a cumulative distribution function after log transformation to avoid fitting negative (unphysical) membrane distances. The log normal cumulative distribution function was fit, using the following model, with Matlab (2019b, The MathWorks, Inc.):

$$f(x) = a*normcdf(x, mu1, sigma1) + (1 - a)*normcdf(x, mu2, sigma2)$$

where $a$ is the relative contribution of each mixture component, mu1 and mu2 represent the means, and sigma1 and sigma 2 the standard deviations of each term, respectively; $x$ is membrane distance and $f(x)$ the cumulative distribution function. The domain of a is between 0 and 1 while the domains of mu1 is between 0 and 3.2 in the log transformed data (i.e. 1 and 24.5 nm back-transformed), and mu2 between 2.7 and 5 (i.e. 15 and 148 nm back-transformed).

**Bootstrapping statistics of histograms.** Assuming the analyzed images are representative of the population, confidence intervals of derived parameters can be determined by bootstrapping[43,44], i.e. randomly assembling new datasets from the existing dataset and calculating statistics of the new bootstrapped datasets. This allows comparison of histogram data within an experiment. The two-component mixture model was fit to each bootstrapped distribution. From the resulting parameters the means and 95% confidence intervals (central 95% of the parameter distribution) were determined.

The data of the PfNCR1 immuno-gold EM was taken from 26 images (of 24 cells) in which 88 gold particles at the PV (18 nm diameter) were identified near (18 nm from the gold surface, i.e. primary and secondary antibody length) recognizable PPM, and with recognizable PVM. All measurable membrane distances in the dataset were recorded. To compare parameter $a_{all}$ of all PVM-PPM distances present in the dataset with $a_{gold}$ of the gold particle distribution, the membrane distance distribution was divided into 54 nm long PV segments (corresponds to the range in which a PfNCR1 protein could be located from the center of a gold particle, i.e. length of 2x primary + 2x secondary antibody + gold), 1118 segments total (60.3 μm PV length) in the dataset, and bootstrapped with replacement 10,000×. The data of all available membrane distances describes all the regions in which a gold particle could be found. Hence, mu1, mu2, sigma1 and sigma2 of the bootstrap calculation of all distances can be used to determine parameter $a_{gold}$ in the gold particle distribution. This is achieved by fixing the parameters to the found values in the bootstrap calculation of distances near gold particles.

The enrichment of PfNCR1 was calculated as $(a_{gold} (1 - a_{all}))/(a_{all} (1 - a_{gold}))$.

Images of regions of single slices of randomly chosen cells were acquired to compare the PfNCR1 knock-down (34 images) with its control (46 images). Region to region differences are assumed to introduce variance in the sampling of membrane distances in electron micrographs. Hence, statistics was calculated by a bootstrap with replacement 10,000× of the images in the individual dataset. The significance of the difference in the relative contribution parameter $a$ was determined by pooling the knock-down with the control dataset and bootstrapping with replacement into two random distributions with the size of the original distributions. This results in 2 a parameters $a_1$ and $a_2$ from the two mixed distributions. The difference of $a_1$ and $a_2$ was used determine the 95% confidence interval for a significant difference in between the knock-down and control dataset.

**EM of immuno-gold labeled PfNCR-GFP and PfNCR1 knock-down.** Samples were prepared for the initial characterization of PfNCR1 in Istvan et al.[10] and reused in this publication. The existing immuno-gold image dataset was thus analyzed. The dataset was imaged with no hypothesis regarding PVM-PPM distance as outlined in[10]. Additional images were acquired of the PfNCR1 knock-down and control sample using a Thermo Fisher Technai T20 transmission electron microscope operated at 200 kV. Images were collected using an AMT NanoSprint1200 (Advanced Microscopy Techniques, Woburn, MA), a CMOS TEM camera.

**Correlative light electron microscopy.** EXP2-mNeonGreen—PV-mRuby3 parasites were isolated on a 65% percoll (MilliporeSigma) interface. Cells were fixed using 4% formaldehyde + 0.4% glutaraldehyde in 1x PHEM (pH 6.9) (Electron microscopy sciences)[45]. Gelatin-embedded samples were infiltrated with 2.3 M sucrose (in 0.1 M phosphate buffer) and put at 4 °C for three days on a rotating wheel, mounted onto sample pins and frozen in liquid nitrogen. Subsequently, the samples were ultrathin cryo-sectioned (50–60 nm) with a FC7/UC7-ultra-microtome (Leica, Vienna, AT) and with a 35° diamond knife (Diatome, Hatfield

PA), picked-up with a 1:1 mixture of 2% methylcellulose (25 centipoises, MilliporeSigma) and 2.3 M sucrose (USB Corporation, Cleveland, OH)[46]. Sections were 5–10 min washed in PBS prior to light microscopy imaging using the Zeiss LSM 880 Airyscan module (see the section on light microscopy). Samples were then embedded in 4% uranyl acetate/2% methylcellulose mixture (ratio 1:9)[47] for electron microscopy. Thin sections were examined in a JEM-1200EX (JEOL USA) transmission electron microscope (accelerating voltage 80 KeV) equipped with an A.M.T. 6-megapixel digital camera (Advanced Microscopy Techniques, Woburn, MA). EC-CLEM[48] was used to align light and electron microscopy image in a non-rigid grid to accommodate sample warping. Electron micrographs with up to ×2500 direct magnification (7.4 nm pixel size) were used for the alignment to the light microscopy image (pixel size following the "optimal" pixel size settings of the Airyscan module is 42.6 nm).

**Freeze fracture replica.** Late stage NF54attb infected red blood cells were isolated using a magnetic separator as for the thin sections. Cell were gently pelleted at ~200 g in a clinical centrifuge, then layered as a thick slurry on a tiny 3 × 3 mm class coverslip mounted on a lung cushion, in preparation for quick-freezing with the liquid helium cooled copper-block slammer[49]. Thereafter, they were transferred to a Balzers 400 freeze-fracture apparatus, where they were fractured through their well-frozen surfaces, deep-etched for 2 min at −104 °C, and then rotary-replicated with 4 nm of platinum deposited from a 20° angle. Thereafter they were backed with 10 nm of carbon deposited from 90°, removed from the Balzers, thawed, and the platinum replica was floated on 25% SDS to partially remove organic material from underneath it[50]. After washing, the replica was picked up on an EM grid. Thereafter, these replicas were examined in the electron microscope in the same manner as the thin sections, above.

**Reporting summary.** Further information on research design is available in the Nature Research Reporting Summary linked to this article.

## Data availability

Data supporting the findings of this manuscript are available from the corresponding authors upon reasonable request. A reporting summary for this Article is available as a Supplementary Information file. The source data underlying histograms, plots, averages with error are provided as a Source Data file. Source data are provided with this paper. An example image is provided with the scripts.

## Code availability

MATLAB (Mathworks) scripts as described in the methods are available at https://github.com/gartenm/piebald.

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

## Acknowledgements

We thank Svetlana Glushakova for critical discussions, Paul Blank for particularly useful discussions and help with the statistical analyses, Jennifer Petersen for help with electron microscopy, Rumiana Dimova for pointing out the refractive index correction, Wandy Beatty for PfNCR1 EM. This work was supported by the Division of Intramural Research of the *Eunice Kennedy Shriver* National Institute of Child Health and Human Development, National Institutes of Health and National Institutes of Health grant HL133453 to J.R.B.

## Author contributions

M.G., J.Z., D.E.G., and J.R.B. conceived and designed experiments. J.R.B. generated and analyzed the parasite strains, performed light microscopy of the DHFR strain, generated the samples for the freeze fracture. M.G. performed all other light microscopy, EM of the PfNCR1 knock-down and control, the IFA, analyzed all microscopy data. R.R. prepared the freeze-fractures. E.S.I. prepared PfNCR1 immuno-gold and knock-down samples. T.T.-H. embedded and cut the thin-sections for EM, J.H. imaged freeze-fractures and thin sections. C.K.E.B. processed the sample for CLEM, imaged the sample in EM. All authors discussed and edited the paper.

## Competing interests

The authors declare no competing interests.
