## [Peer Review File · Nature Communications]

Peer Review File - Reviewers' comments first round:

Reviewer #1 (Remarks to the Author):

The manuscript by Garten et al builds upon several previous observations. These include: the parasitophorous vacuole membrane (PVM) proteins such EXP2 (a component of PTEX) display a 'beads on a string'-like appearance, and that interfering with PTEX function in a manner that leads to a block in protein export across the PVM results in an expansion of these proteins in the vacuolar space in a non-homogenous manner. These studies all point to the proteins localizing to particular domains as a result of particular adhesion points existing between the parasite membrane (PM) and PVM. It has also been shown by Garten in a previous manuscript that PTEX components at the PVM are freely accessible to vacuolar proteins.

In this manuscript, Garten et al use parasites expressing EXP2-mNeon Green to show EXP2 localises in domains of the PVM and that EXP2 co-localises with the membrane lumen, confirming previous observations above. The unique aspect of this paper is the use of correlative light EM to show that in regions where the PM and PVM are closely apposed, the ruby-labelled PV protein is absent, and by Mollweide projections of EXP2 and a PV protein, that these two proteins co-localise. The authors then go on to show that the lipid transport protein PfNCR1 does not co-localise with EXP2 or the PV protein - from this they conclude in their model that PfNCR localises to regions in which the PPM and PVM are closely apposed, although the authors did not actually demonstrate this via correlative light EM as per Fig 3A. Hence the segregation of function (eg. protein and small molecule transport versus lipid transport) is most likely mediated by the creation of membrane contact sites, although such sites could not be observed. To strengthen the conclusions it would have been good to confirm results with other PVM resident proteins that are not involved in lipid transport.

On Line 183-186: The authors mention that the regions of close membrane apposition are a promising place to look for a machinery that would allow transfer of proteins from the PPM to the PVM. Do the authors mean proteins inserted into the PVM or that then go through PTEX. Several studies have shown that at least some proteins containing a TM domain (both PEXEL and PEXEL-negative proteins) require an unfolding step at the PPM prior to accessing PTEX at the PVM, in which case would one not expect the PPM machinery extruding out PPM proteins into the vacuolar space (or directly onto PTEX) where the membranes are not closely apposed?

Statistical analysis needs to be performed for Fig. 2F.

Reviewer #2 (Remarks to the Author):

Garten et al. describe membrane contact sites between the parasite plasma membrane and the PVM in *Plasmodium falciparum*-infected erythrocytes. They used cutting edge microscopy techniques to identify and characterize MCS very convincingly.

I have no major concerns, just a few points that need clarification.

Figure 3: the CLEM image is very convincing but I was wondering how many examples of this correlation they have examined. Could this CLEM experiment be quantified or could they at least provide more examples in the supplementary section?

They might consider presenting a single confocal slice next to the Mollweide projections for Exp2-mNG and PV-mRuby3. This would show the negative correlation very precisely and support their hypothesis even more.

Line 95-110: They suggest co-localization of Exp2 and PV-Ruby3 because light microscopy does not allow resolution of 20 nm, which is about the distance of the 2 membranes separated by the PV lumen. However, technically this is not correct as Exp2 is restricted to the PVM and PV-Ruby3 to the lumen of the PV. They should point this out very clearly to not mislead the reader. Cell fractioning and western blotting would support the different localization of Exp2 and PV-Ruby3.

The anti-correlation of PfNCR1 and Exp2 localization with -0,2 is less convincing considering that -1 would be perfect anti-correlation. They need to comment on this, in particular since in a considerable number of infected cells, the correlation is even positive (figure 4B). I assume that it has a biological meaning that sometimes there is a very strong anti-correlation and sometimes PfNCR1 and Exp2 localization correlates even positively. Perhaps different time points of development like early and late trophozoites have been imaged. Highly synchronous parasite cultures might help to address this. They should discuss these facts and possible options in more detail.

Since they don't show any transport mechanism, they should tone down their theory of hydrophilic and hydrophobic transport. I like this idea but it certainly needs mechanistic evidence to make such a statement.

Line 177: what kind of processes in the parasite cytoplasm do they have in mind? Please explain.

Contacting domains segregate a lipid transporter from a solute transporter in the malarial host-parasite interface

Matthias Garten¹, Josh R. Beck³, Robyn Roth⁴, Tatyana Tenkova-Heuser¹, John Heuser¹, Eva S. Istvan⁴, Christopher K. E. Bleck², Daniel E. Goldberg^{4*}, and Joshua Zimmerberg^{1*}

Response to Reviewers:

We thank the reviewers profusely for their insightful comments; the resulting manuscript is much improved. Also, during the review process we were told by a colleague of another improvement: a refractive index correction to the projection algorithm. The changes in the projections are minor to our eyes, but correcting for refractive index improved the precision of the analyzed results. Therefore, the revised manuscript has recalculated numerical values based on the projections. No conclusions have changed by dint of this recalculation.

In accessing the samples and datasets created for Istvan et al. 2019 to localize PfNCR1 and answer reviewer 1's major concern, we were able to ask if PfNCR1 is an important contributor to the structure of the PV. This was found to be a useful addition to the manuscript. See Figure SI 5 for the addition.

Minor changes:

- 1) Histograms showing PVM-PPM distances are now scaled on the y-axis. This gives information on the length of measured PV.
- 2) The histogram in figure 1 was fit to the two-component mixture model.
- 3) While double checking all numbers in the manuscript for the resubmission, we found that the Pearson coefficients in the Figure Legends (but not anywhere else) were obtained after thresholding and not without thresholding as intended. This is now corrected. No conclusions were affected.

Reviewer #1 (Remarks to the Author):

The manuscript by Garten et al builds upon several previous observations. These include: the parasitophorous vacuole membrane (PVM) proteins such EXP2 (a component of PTEX) display a 'beads on a string'-like appearance, and that interfering with PTEX function in a manner that leads to a block in protein export across the PVM results in an expansion of these proteins in the vacuolar space in a non-homogenous manner. These studies all point to the proteins localizing to particular domains as a result of particular adhesion points existing between the parasite membrane (PM) and PVM. It has also been shown by Garten in a previous manuscript that PTEX components at the PVM are freely accessible to vacuolar proteins.

In this manuscript, Garten et al use parasites expressing EXP2-mNeon Green to show EXP2 localises in domains of the PVM and that EXP2 co-localises with the membrane lumen, confirming previous observations above. The unique aspect of this paper is the use of correlative light EM to show that in regions where the PM and PVM are closely apposed, the ruby-labelled PV protein is absent, and by Mollweide projections of EXP2 and a PV protein, that these two proteins co-localise.

The authors then go on to show that the lipid transport protein PfNCR1 does not co-localise with EXP2 or the PV protein - from this they conclude in their model that PfNCR localises to regions in which the

PPM and PVM are closely apposed, although the authors did not actually demonstrate this via correlative light EM as per Fig 3A.

We agree with the reviewer that in the original manuscript PfNCR1 localization is deduced only from its anti-correlation with EXP2 and PV-mRuby3. As this is a major concern for the publication, we prepared a correlative light EM sample of the PfNCR1-GFP—EXP2-mRuby3 parasite line. Unfortunately, the fluorescent background originating from the support and the fixation of these new Tokuyasu sections elevated the noise in the GFP channel over the signal, preventing detection of PfNCR1-GFP (see 'Response Figure 1' below). In the EM it was found that regions showing some candidate GFP signal were either a fold in the sample or showed an obliquely cut parasite making identification of the membrane impossible. Compared to the mNeonGreen signal, the GFP signal is too weak to be directly detected in the sections. This is probably due to GFP being a less efficient fluorophore compared to mNeonGreen and endogenous PfNCR1 expression levels being substantially lower than those of EXP2 (8.5x lower as determined by the average of the ratios of the geometric mean expression in Otto et al. *MolMicrobiol* 2010 PMID: 20141604).

Thus, to visualize PfNCR1 its localization had to be enhanced with an immuno EM technique. Such a sample was already prepared for the work in Istvan et al. 2019 for the initial characterization of PfNCR1 showing localization of PfNCR1-GFP with the help of an immuno-gold label. We thus reanalyzed the previously acquired dataset. This was ideal in one sense -- the experimenter acquiring the images years ago was truly blinded to the current hypothesis. On the other hand, immuno-gold labeled EM samples lack the continuous fluorescent label seen in CLEM due to the sparse nature of the immuno-gold binding, requiring statistical analysis of immuno-gold label to achieve a comparable rigor in the conclusion. We analyzed all gold particles in the dataset that were in close proximity (i.e. 18 nm from the gold surface [approximately combined primary and secondary antibody length]) to the PVM, indicating specific labeling at the PV, that also showed recognizable PVM and PPM at the parasite periphery. To compare the result of the gold label all recognizable PVM – PPM distances of the dataset were recorded. Plotting both histograms together gives a strong sense of enrichment of the label at sites of closer membrane apposition (Figure SI 4). To validate this impression using statistical tools, both distributions (membrane distance at the gold particle and all membrane distances) were then compared using "bootstrapping" analysis. With that we found that PfNCR1 is enriched in regions of close apposition by a factor of 40.0x [14.6x, 61.4x] (mean [95% CI]). Together with the correlation analysis of the fluorescence this result greatly strengthens the confidence in the localization of PfNCR1 at sites of close membrane apposition and we hope it satisfies the reviewer's request.

'Response Figure 1'. Fluorescent image of a Tokuyasu section of the PfNCR1-GFP—EXP2-mRuby3 parasite line. Left to right: PfNCR1-GFP (green), EXP2-mRuby3 (red), merge.

Hence the segregation of function (eg. protein and small molecule transport versus lipid transport) is most likely mediated by the creation of membrane contact sites, although such sites could not be observed. To strengthen the conclusions it would have been good to confirm results with other PVM resident proteins that are not involved in lipid transport.

We agree with the reviewer's constructive remark. Information about the relative colocalization will be useful for other proteins at the host-parasite interface. Indeed, the literature is rich with publications localizing PV proteins (e.g. Spielmann et al. MBoC 2002 PMID: 12686607, Spielmann et al. MolMicrobio 2006 PMID: 16420351, Riglar et al. NatComm 2013 PMID: 23361006, Batinovic et al. NatComm 2017 PMID: 28691708). Most PVM proteins are not functionally characterized. Only recently EXP1 was characterized as important for targeting and function of EXP2 (Mesén-Ramírez et al. 2019 PLOS Biol PMID: 31568532 and Nessel et al. 2020 PMID: 31990132). In Mesén-Ramírez et al. Figure S5 and Riglar et al. 2013, EXP1 and EXP2 colocalize in IFAs. We added two sentences in the discussion: "Recently, EXP1, a PVM protein that colocalizes with EXP2 (Riglar et al. 2013, Mesén-Ramírez et al. 2019), has been shown as important for peripheral EXP2 localization around the parasite and function of the EXP2 as nutrient-permeable channel (Mesén-Ramírez et al. 2019, Nessel et al. 2020). It remains to be seen how EXP1 is affecting EXP2 localization and function."

Co-localization experiments at the host-parasite interface will be useful for classifying the set of proteins at the tight and loose host cell-parasite interface, whether a protein could be involved in solute or lipid transport, or maybe involved in structure generation of the PV, so we advocate for both structural localization of other proteins as well as investigation into their function. In other words, we prefer to reserve localization of PVM and PPM proteins for follow up studies.

On Line 183-186: The authors mention that the regions of close membrane apposition are a promising place to look for a machinery that would allow transfer of proteins from the PPM to the PVM. Do the authors mean proteins inserted into the PVM or that then go through PTEX. Several studies have shown that at least some proteins containing a TM domain (both PEXEL and PEXEL-negative proteins) require an unfolding step at the PPM prior to accessing PTEX at the PVM, in which case would one not expect

the PPM machinery extruding out PPM proteins into the vacuolar space (or directly onto PTEX) where the membranes are not closely apposed?

The reviewer is absolutely correct. We meant to write “PVM resident proteins” as referenced by the Tribensky et al. paper. We removed the Matthews et al. reference for clarity. The sentence is now corrected.

Statistical analysis needs to be performed for Fig. 2F.

We thank the reviewer for the remark. The data in Figure 2F shows a statistically significant difference ($P < 0.0001$). However, several points make a statistical comparison problematic: the culture is not synchronized to make sure similar amounts of EXP2 is expressed, the stains are slightly different (NF54 from the repository vs NF54attb after two rounds of engineering), the technique is different (immune fluorescence assay vs live microscopy), the percent coverage is obtained by thresholding and even though it is done by an algorithm the different sample preparation techniques changes the image and influences the thresholding. As we have no practical way to separately estimate the impact of each of these variables to the overall uncertainty of the findings, panel 2F was meant to make the qualitative point that there are two different areas (One area with abundant protein and one with scarce protein) that can be visualized with light microscopy in both samples on a number of cells. Since that main point is already made with Figure 2a, b, d, and e, for clarity of the main figure we moved panel F to the supplement and plotted both distributions on separate axes.

Reviewer #2 (Remarks to the Author):

Garten et al. describe membrane contact sites between the parasite plasma membrane and the PVM in Plasmodium falciparum-infected erythrocytes. They used cutting edge microscopy techniques to identify and characterize MCS very convincingly.

I have no major concerns, just a few points that need clarification.

Figure 3: the CLEM image is very convincing but I was wondering how many examples of this correlation they have examined. Could this CLEM experiment be quantified or could they at least provide more examples in the supplementary section?

We agree with the reviewer that showing only an example is unsatisfactory. Due to the difference in resolution between the light and electron microscopy quantification is challenging. On both microscopes data is acquired separately, with relatively few high magnification EM images to resolve PV structure in areas that show interesting fluorescence (here an interrupted line of intense fluorescence). The merge of both techniques is done offline using “low” resolution EM and light microscopy. Low-resolution and high-resolution data is presented side by side in the figure panels to zoom into regions that were chosen as interesting while taking the EM images. The supplementary CLEM figure shows now two more high magnification zooms into the fluorescence image to address the concern raised here.

Additionally, in response to reviewer 1’s concern about the PfNCR1 localization, we localized the immuno-gold label relative to the two PV regions. With that we have now two techniques directly showing protein localization follows PV structure. We hope our updated presentation is satisfactory.

They might consider presenting a single confocal slice next to the Mollwied projections for Exp2-mNG

and PV-mRuby3. This would show the negative correlation very precisely and support their hypothesis even more.

We thank the reviewer for the suggestion. In the time of the revision we as well realized that the projections would profit from a side-by-side presentation with a more classical image. We are now showing confocal slices of the dataset with all the projections. We hope this enhances the readability of the figures.

Line 95-110: They suggest co-localization of Exp2 and PV-Ruby3 because light microscopy does not allow resolution of 20 nm, which is about the distance of the 2 membranes separated by the PV lumen. However, technically this is not correct as Exp2 is restricted to the PVM and PV-Ruby3 to the lumen of the PV. They should point this out very clearly to not mislead the reader. Cell fractioning and western blotting would support the different localization of Exp2 and PV-Ruby3.

We thank the reviewer for pointing out that the relative colocalization of PV-mRuby3 and EXP2 is not rationalized very well. A sentence clarifying the localization and colocalization within the limits of fluorescence microscopy was added at line 108: "EXP2 is bound to the PVM, while PV-mRuby3 is detected in the PV-lumen in between the PVM and PPM, on average 20-40 nm from the PVM." In later stages, at the time of egress PV-mRuby3 can clearly be seen to be soluble. It is found in the newly developed space between the forming merozoites and leaks into the red blood cell cytosol the moment the PVM ruptures, while EXP2 remains in the slowly vesiculating membrane (see Figure 3a of the already referenced paper Glushakova et al. 2018).

A saponin release and cell fractionation experiment was done in Nessel et al. 2020 PMID: 31990132. In Figure S4 of Nessel et al. PV-mRuby3 is saponin released while EXP2 remains in the pellet. We now reference this paper with the introduction of PV-mRuby3 to this work.

The anti-correlation of PfNCR1 and Exp2 localization with -0,2 is less convincing considering that -1 would be perfect anti-correlation. They need to comment on this, in particular since in a considerable number of infected cells, the correlation is even positive (figure 4B). I assume that it has a biological meaning that sometimes there is a very strong anti-correlation and sometimes PfNCR1 and Exp2 localization correlates even positively. Perhaps different time points of development like early and late trophozoites have been imaged. Highly synchronous parasite cultures might help to address this. They should discuss these facts and possible options in more detail.

We thank the reviewer for the constructive remark. Our interpretation is that the positive values are a combination of limited optical resolution of the light microscope and the wide size distribution of the domains. Basically, cells in which domains are close in size to the optical resolution limit will have a relatively large contribution of the border areas to the total computed correlation value compared to cells with domains much larger than optical resolution. At a domain border both signals are present and are thus co-localized.

To illustrate the dynamic nature of the domain size over time we added a movie (SI movie 2) to this manuscript that shows an entire lifecycle of a parasite with labeled EXP2-mNeonGreen and another translocon component (PTEX150-mRuby3). Our general impression is that the EXP2 signal is more punctate earlier, forming larger domains later. A detailed exploration of the EXP2 distribution over all the stages of the erythrocytic lifecycle is out of the limited scope of this manuscript that focuses on trophozoite stage parasites.

To better address the resolution limit, we expanded the explanation in line 129 to read:
“Positive values of the coefficient are caused by small domains, approaching the resolution limit of light microscopy (~130nm in x-y direction and ~400 nm in z) with signals co-localizing at the border of domains, in contrast, larger domains having fewer domain borders have more negative coefficients (see the sequence in Figure 4A left to right).”

Since they don't show any transport mechanism, they should tone down their theory of hydrophilic and hydrophobic transport. I like this idea but it certainly needs mechanistic evidence to make such a statement.

We agree and thank the reviewer for pointing out the statement: we do not measure transport, only proteins that are transporters. In addition, it is only with 2 transporters. It is not yet a theory, just a proposal. Future studies will have to test the generality of this hypothesis. Thus, we have revised the discussion:

“Taking all this data into account, we posit that the PV has evolved to become laterally segregated into regions for hydrophilic transport, and separate closer-contact regions for hydrophobic transport. Furthermore, we have altered the title to “Contacting domains segregate a lipid transporter from a solute transporter in the malarial host-parasite interface”.

Line 177: what kind of processes in the parasite cytoplasm do they have in mind? Please explain.

We thank the reviewer for pointing out the vague sentence. We changed the sentence to be more specific (line 192).

“Curiously, the domain structure of the PVM exemplified by the EXP2 distribution shown here, is quite dynamic and variable (*cf.*, SI movie, demonstrating remarkable flexibility in the PV), suggesting active mechanisms of protein localization driven by active processes, e.g. cytoskeletal rearrangements coupled to the PVM by contact sites, or on-going exocytosis and solute export modifying the PV lumen.”

Peer Review File - Reviewers' comments second round:

REVIEWERS' COMMENTS:

Reviewer #1 (Remarks to the Author):

In the original manuscript the authors deduced PfNCR1 localization from its anti-correlation with EXP2 and PV-mRuby3. In the revised manuscript, the authors resorted to an immune-EM technique to validate that the PfNCR1 immunolabelling was near the sites of close parasite membrane-parasite vacuolar membrane opposition (Fig S1.4). To be honest I couldn't understand panel F and how that was really derived. They then go onto show that whilst PfNCR1 localises to these regions, it is not critical for the close apposition of the PM and PVM.

The caveat of the study is that the analysis has been performed on just one lipid transporter and one protein involved in small molecule/protein export, it is unclear whether the creation of membrane contact sites is truly representative of segregation of function. Consequently, I think in even the revised version that the statement 'we posit that the PV has evolved to become laterally segregated into regions for hydrophilic transport, and separate closer-contact regions for hydrophobic transport' is still a big call.

Reviewer #2 (Remarks to the Author):

Thank you for the very detailed and professional response to my few minor concerns. All have been addressed very convincingly and I have no further concerns. Congratulation for a very nice study!

We are thankful for the reviewers' work and the comments we received for our revised manuscript. We thank the reviewers for the valuable contributions made to our work. Please find the remaining comments addressed below.

Reviewer #1 (Remarks to the Author):

In the original manuscript the authors deduced PfNCR1 localization from its anti-correlation with EXP2 and PV-mRuby3. In the revised manuscript, the authors resorted to an immune-EM technique to validate that the PfNCR1 immunolabelling was near the sites of close parasite membrane-parasite vacuolar membrane opposition (Fig S1.4). To be honest I couldn't understand panel F and how that was really derived.

We thank the reviewer for critically reading the figure legend. We revised the legend of panel F and hope that we were able to improve comprehensibility of the legend.

They then go onto show that whilst PfNCR1 localises to these regions, it is not critical for the close apposition of the PM and PVM.

The caveat of the study is that the analysis has been performed on just one lipid transporter and one protein involved in small molecule/protein export, it is unclear whether the creation of membrane contact sites is truly representative of segregation of function. Consequently, I think in even the revised version that the statement 'we posit that the PV has evolved to become laterally segregated into regions for hydrophilic transport, and separate closer-contact regions for hydrophobic transport' is still a big call.

We agree that the statement can be formulated more carefully. We now propose the hypothesis more cautiously and added a sentence that further evidence in support of the hypothesis needs to be collected.

"Taking all this data into account, we propose that the PV has evolved to become laterally segregated into regions for hydrophilic transport, and separate closer-contact regions for hydrophobic transport. It will be necessary to functionally characterize and localize other proteins at the HPI to further bolster this hypothesis."

Reviewer #2 (Remarks to the Author):

Thank you for the very detailed and professional response to my few minor concerns. All have been addressed very convincingly and I have no further concerns. Congratulations for a very nice study!

We are happy to read that we have fulfilled all of reviewer 2's requests to his/her full satisfaction.